# Unlabeled Disentangling of GANs with Guided Siamese Networks

## Abstract

Disentangling underlying generative factors of a data distribution is important for interpretability and generalizable representations. In this paper, we introduce two novel disentangling methods. Our first method, Unlabeled Disentangling GAN (UD-GAN, unsupervised), decomposes the latent noise by generating similar/dissimilar image pairs and it learns a distance metric on these pairs with siamese networks and a contrastive loss. This pairwise approach provides consistent representations for similar data points. Our second method (UD-GAN-G, weakly supervised) modifies the UD-GAN with user-defined *guidance* functions, which restrict the information that goes into the siamese networks. This constraint helps UD-GAN-G to focus on the desired semantic variations in the data. We show that both our methods outperform existing unsupervised approaches in quantitative metrics that measure semantic accuracy of the learned representations. In addition, we illustrate that simple guidance functions we use in UD-GAN-G allow us to directly capture the desired variations in the data.

## 1 Introduction

Generative Adversarial Networks (GANs) (Goodfellow et al., 2014) are generative model estimators, where two neural networks (generator and discriminator) are trained in an adversarial setting, so that likelihood-based probabilistic modeling is not necessary. This works particularly well for sampling from a complex probability distribution, such as images. Although GANs yield realistic looking images (Radford et al., 2015), the original formulation in (Goodfellow et al., 2014) only allows for randomly sampling from the data distribution without disentangled structural or semantic control over the generated data points.

One way to disentangle the generation process is to use conditional GANs (Mirza & Osindero, 2014; Odena et al., 2017). These models modify the generator by conditioning it with supervised labels. Then, they either take the same labels as input in the discriminator (Mirza & Osindero, 2014) and measure the image-label compatibility, or classify the correct label at the output, given the generated image (Odena et al., 2017). Conditional GANs rely on a dataset with labels, which might not always be available or might be time-consuming to collect.

In this paper, we propose two GAN-based methods that learns disentangled representations without using labeled data. Our first method, Unlabeled Disentangling GAN (UD-GAN), generates image pairs, then embeds them with Siamese Networks (Chopra et al., 2005), and finally learns a distance metric on a disentangled representation space. Whereas our second method, UD-GAN-G, uses guidance functions to restrict the input to our siamese networks, so that they capture desired semantic variations.

## 2 Related Work

There have been many studies on learning disentangled representations in generative models, which can be grouped into the level of supervision/labeled data they require.

**Disentangled representations (supervised).** In (Zhu et al., 2014; Yang et al., 2015), the identity and the viewpoint of an object are disentangled via reconstructing the same object from a different

viewpoint and minimizing a reconstruction loss. Whereas in (Kingma et al., 2014; Makhzani et al., 2016), the style and category of an object is separated via autoencoders, where an encoder embeds the style of an input image to a latent representation, and a decoder takes the category and style input to reconstruct the input image. In (Tran et al., 2017; Yin et al., 2017), autoencoders and GANs are combined to decompose identity and attribute of an object, where the disentangled representation is obtained at the encoder outputs, and image labels are used at the output of the discriminator.

**Disentangled representations (semi-supervised).**   In (Reed et al., 2014), they clamp the hidden units for a pair of images with the same identity but with different pose or expression to have the same identity representation. Whereas in (Kulkarni et al., 2015), synthesized images are used to disentangle pose, light, and shape of an object by passing a batch of images where only one attribute varies and the rest of the representation is clamped to be the same. These techniques only require a batch of samples with one attribute different at a time.

**Disentangled representations (unsupervised).**   InfoGAN (Chen et al., 2016) is an unsupervised technique that discovers categorical and continuous factors by maximizing the mutual information between a GAN's noise variables and the generated image. $\beta$-VAE (Higgins et al., 2017) and DIP-VAE (Kumar et al., 2018) are unsupervised autoencoder-based techniques that disentangle different factors in the latent representation of an encoded image. In $\beta$-VAE, the KL-divergence between the latent and a prior distribution is weighted with a factor $\beta > 1$ to encourage disentanglement in the posterior latent distributions. Wheres in DIP-VAE, the covariance matrix of the latent distribution is encouraged to be an identity matrix, thus leading to uncorrelated latent representations.

For all of the unsupervised methods, after a model is trained, a human needs to investigate which factors map to which semantic property. In addition, as the methods are unsupervised, not all desirable factors might be represented. In contrast, our method builds on existing approaches with two important modifications: (i) We operate on pairs of similar/dissimilar image pairs. (ii) We compute the image embeddings using separate networks, which allows us to guide the disentangling process with information restriction.

# 3   UNLABELED DISENTANGLING GAN

## 3.1   BACKGROUND: GENERATIVE ADVERSARIAL NETWORKS

In GANs, the generator, $G(.)$, maps a latent variable $\mathbf{z}$, which has an easy-to-sample distribution, into a more complex and unknown distribution, such as images. On the other hand, the discriminator $D(.)$ tries to distinguish real images from the ones that are generated by $G$. In (Goodfellow et al., 2014), the training is performed as a minimax game as follows:

$$\min_G \max_D V(G, D) = \mathbb{E}_{\mathbf{x} \sim \mathbb{P}_\mathrm{R}} [\log D(\mathbf{x})] + \mathbb{E}_{\mathbf{z} \sim \mathbb{P}_\mathrm{Z}} [\log(1 - D(G(\mathbf{z})))], \tag{1}$$

where $\mathbb{P}_\mathrm{R}$ and $\mathbb{P}_\mathrm{Z}$ are the probability distributions of real images and the latent variable $\mathbf{z}$, respectively. We train our GAN by using the loss in equation 1. In order to increase stability, we modify the generator loss by maximzing $\log(D(G(\mathbf{z})))$, instead of minimizing the second term in equation 1.

## 3.2   A NOVEL GAN ARCHITECTURE: UD-GAN

In a standard GAN setting, all of the variation in the distribution of real images is captured by the latent variable $\mathbf{z}$. However, a single dimension or a slice of $\mathbf{z}$ does not necessarily have a semantic meaning. In this paper, our target is to slice the latent variable into multiple vectors, where each vector controls a different semantic variation.

Our network architecture is visualized in Figure 1. In our method, the latent vector $\mathbf{z} = [\mathbf{q}_1, \mathbf{q}_2, ..., \mathbf{q}_{N_A}]$ is the concatenation of $N_A$ *knobs*, $\{\mathbf{q}_i\}_{i=1}^{N_A}$, which represent different attributes we aim to disentangle. One can add a final variable that captures the variation (and the noise) that is not picked up by the knobs. In our experiments, this additional variable did not have a notable effect. In our notation, $\mathbf{q}_{\bar{i}}$ refers to all of the knobs, except $\mathbf{q}_i$. In order to train our model, first, for each $\mathbf{q}_i$, we sample two different vectors, $\mathbf{q}_i^{(1)}$ and $\mathbf{q}_i^{(2)}$ from $\mathtt{Unif}(-1, 1)$. If we would attempt to form a batch by combinatorially concatenating all knob samples, we get a batch size of $2^{N_A}$, which

grows exponentially with the number of attributes. To avoid this computational burden, we train our model through stochastic sampling of one attribute at a time. For example, if the $i^{th}$ attribute is chosen, we generate four images as shown in Figure 1.

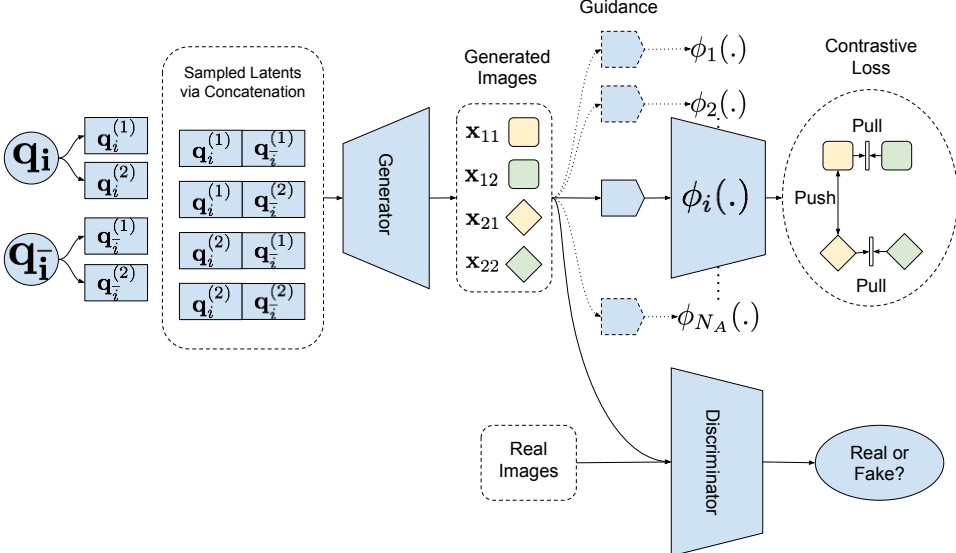

Figure 1: The flowchart of our architecture. Sampled latents from different attributes are combined into latent vectors. Generated images are grouped with respect to different attributes (here, represented by shape) by Siamese Networks (denoted as $\phi_i$).

The image pairs that are generated with the same $\mathbf{q}_i$ vectors, $\{\mathbf{x}_{11}, \mathbf{x}_{12}\}$ or $\{\mathbf{x}_{21}, \mathbf{x}_{22}\}$, should have the same $i^{th}$ attribute, regardless of the values of $\mathbf{q}_{\bar{i}}$. We can ensure this via embedding the generated image pairs into a representation space with Siamese Networks (Chopra et al., 2005), which are denoted as $\phi_i(.)$, and then learning a distance metric on the embedding vectors by employing Contrastive Loss (Hadsell et al., 2006). An optional *guidance* function is used to restrict the information that goes into a siamese network, thus letting us approximate a desired representation space. The guidance is disabled for our unsupervised UD-GAN approach. Whereas for UD-GAN-G, the guidance is a simple, user-defined function, which is discussed in Section 3.3.

We use a Contrastive Loss function to pull similar image pairs together, and push dissimilar pairs apart as follows:

$$\mathcal{L}_{\phi_i} = \frac{1}{2} \sum_{n_i=1}^{2} \rho_i(\mathbf{x}_{n_i 1}, \mathbf{x}_{n_i 2})^2 + \frac{1}{4} \sum_{n_{\bar{i}}=1}^{2} \sum_{m_{\bar{i}}=1}^{2} \max(0, \gamma_i^{(1,2)} - \rho_i(\mathbf{x}_{1 n_{\bar{i}}}, \mathbf{x}_{2 m_{\bar{i}}}))^2, \qquad (2)$$

where, $\mathcal{L}_{\phi_i}$ is the Contrastive Loss for the $i^{th}$ Siamese Network $\phi_i(.)$, the function $\rho_i(\mathbf{x}_{n_i 1}, \mathbf{x}_{n_i 2}) = \left\| \phi_i(\mathbf{x}_{n_i 1}) - \phi_i(\mathbf{x}_{n_i 2}) \right\|_2$ is a shorthand for embedding distance between $\mathbf{x}_{n_i 1}$ and $\mathbf{x}_{n_i 2}$, and $\gamma_i^{(1,2)}$ is an adaptive margin of the form $\gamma_i^{(1,2)} = \left\| \mathbf{q}_i^{(1)} - \mathbf{q}_i^{(2)} \right\|_2$. Using an adaptive margin makes the distance between two latent samples semantically meaningful and we empirically found that it improves the training stability.

The discriminator network $D$ is not modified and is trained to separate real and generated image distributions. Donahue et al. (2018) use a similar latent variable slicing for capturing illumination and pose variations of a face with a fixed identity. Their discriminator needs image pairs, which must be labeled for real images, to judge the quality and identity of the faces. Our method does not require any labels for the real images. Instead, we create similar and dissimilar image pairs via concatenating latent variables and generating image batches. Our final loss function is:

$$\mathcal{L}_{\phi} = \lambda_{\phi_i} \mathcal{L}_{\phi_i}, \quad i \sim \text{Cat}(N_A)$$
$$\min_{G} \max_{D} V(G, D) = \mathcal{L}_{GAN} + \mathcal{L}_{\phi}, \qquad (3)$$

where, $\mathcal{L}_{WGAN}$ is the GAN loss described in equation 1, $\lambda_{\phi_i}$ is the weight of the embedding loss, and the sampling of the latent variables depends on $i$ and is performed as described above.

## 3.3 DISENTANGLING WITH GUIDANCE FUNCTIONS: UD-GAN-G

A guidance function reduces the information content that flows into a siamese network and causes the corresponding embedding space to capture only the variations in the restricted input. For example, consider we want to capture the hair-related attributes in the CelebA dataset (Liu et al., 2015), which contains aligned images of human faces. By cropping every region but the top part of a generated image, we are able to guide $\phi_{top}(.)$ to learn only the variations in the "Hair Color" as shown in the first row of Figure 2. Note that, the knob $\mathbf{q}_{top}$ (that corresponds to $\phi_{top}(.)$) changes the hair color not only at the cropped part of the image but as a whole. This is due to the interplay between the adversarial part of our loss (see equation 3), which enforces global realism in images, and the contrastive loss, which administers disentangled representations. As shown in Figure 2, different guidance functions leads to capturing different variations in the CelebA dataset.

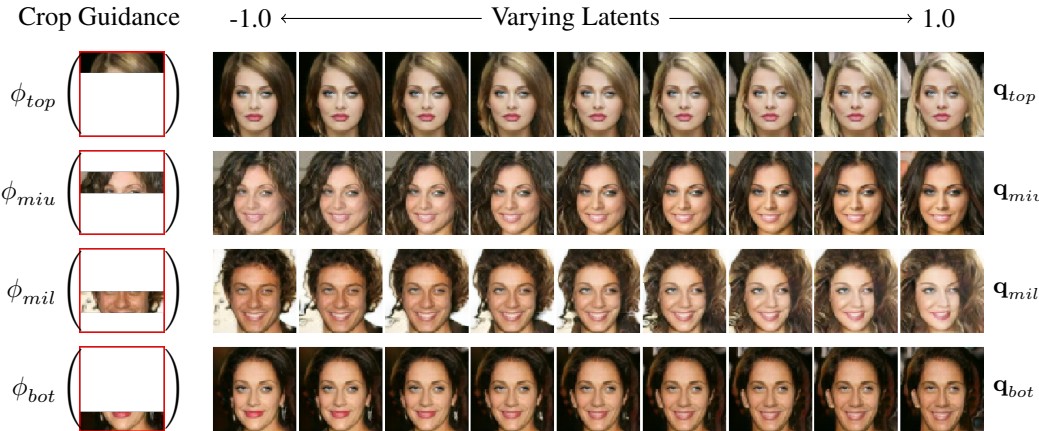

Figure 2: (left) Four of our siamese networks are guided with differently cropped images. (right) Varying latent variables that correspond to guided siamese networks captures desired variations.

## 3.4 PROBABILISTIC INTERPRETATION

We can gain a probabilistic interpretation of our method on a toy example. Let us assume a problem, where we want to generate images of colored polygons (see Figure 1), where there are two independent factors of variation: shape and color, which we want to capture using two knobs $\mathbf{q}_i$ and $\mathbf{q}_j$, respectively. When we set $\mathbf{q}_j$ to a certain value and vary $\mathbf{q}_i$, we want to generate polygons with the same color, but different shapes, and vice versa.

Let $\mathbb{P}$ be the probability distribution of colored polygons. For each attribute, $\mathbb{P}$ can be decomposed into a mixture distribution as follows:

$$\mathbb{P} = \sum_{k=1}^{N_i} \pi_i^{(k)} \mathbb{Q}_i^{(k)} \quad \leftarrow \text{ for attribute } i, \qquad \mathbb{P} = \sum_{k=1}^{N_j} \pi_j^{(k)} \mathbb{Q}_j^{(k)} \quad \leftarrow \text{ for attribute } j \quad (4)$$

where, $\mathbb{Q}_i^{(k)}$ is a mixture component and $\pi_i^{(k)}$ is its corresponding probability of choosing it, and $N_i$ is the number of different values an attribute (in our example, $i$ corresponds to shape) can take. A similar explanation can be made for attribute $j$, i.e. color. For the sake of this analysis, we accept that for each attribute, $\mathbb{P}$ can be decomposed into different discrete mixture distributions as shown in Figure 3. For this specific case, $\mathbb{Q}_i^{(1)}$ and $\mathbb{Q}_i^{(2)}$ are the distributions of colored squares and colored diamonds, respectively. For the color attribute, which is indexed by $j$, each $\mathbb{Q}_j^{(k)}$ corresponds to a distribution of polygons with a single color (i.e., *green polygons*).

Our contrastive loss in equation 2 has two terms. The first term is minimizing the *spread* of each mixture component $\mathbb{Q}_i^{(k)}$. This spread is inversely related to disentanglement. If all samples from

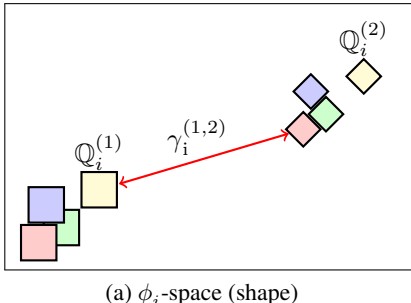 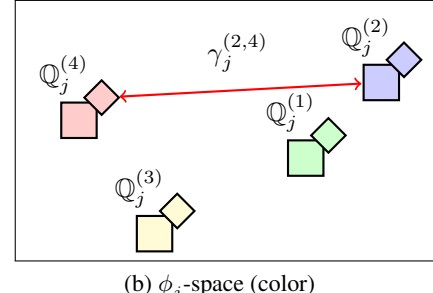

(a) $\phi_i$-space (shape)          (b) $\phi_j$-space (color)

Figure 3: Illustration of the embedding spaces and separated probability distributions after training our model.

$\mathbb{Q}_i^{(k)}$ are mapped to the same embedding vector, the effect of $j$ (and any other attribute) on the representation $\phi_i(.)$ disappears and *disentangling* is achieved. During training, we stochastically go through all embedding spaces and minimize their spread, thus resulting in a disentangled representation in Table 9 in Appendix G.

The second term in equation 2 separates all $\mathbb{Q}_i^{(k)}$ from each other using an adaptive margin $\gamma_i^{(1,2)}$. This margin depends on the difference between input latent pairs, so that the resulting embedding space is smooth. In other words, we separate rectangles, circles, and ovals from each other, but circles should be closer to ovals than squares, due to their relative similarity. In the following, we focus on the shape attribute that is represented by $i$, however, derivations carry over to the color attribute $j$.

In order to separate the probability distributions over image embeddings, one can maximize a divergence between all pairs from $\mathbb{Q}_i^{(k)}$. One way to measure the distance between these distributions is to use the unbiased estimator of the energy distance (Székely & Rizzo, 2004):

$$D_E(\mathbb{Q}_i^{(1)}, \mathbb{Q}_i^{(2)}; \phi_i, j) = -\frac{1}{2}\sum_{n_i=1}^{2}\rho_i(\mathbf{x}_{n_i 1}, \mathbf{x}_{n_i 2})^2 + \frac{1}{4}\sum_{n_j=1}^{2}\sum_{m_j=1}^{2}\rho_i(\mathbf{x}_{1n_j}, \mathbf{x}_{2m_j})^2 \qquad (5)$$

The energy distance in equation 5 can be interpreted as an instance of Maximum Mean Discrepancy (Bińkowski et al., 2018) and resembles the Contrastive Loss (Hadsell et al., 2006). We can rewrite equation 5 using the Contrastive Loss in equation 2 as follows:

$$D_E = -\mathcal{L}_{\phi_i} + \frac{1}{4}\sum_{n_j=1}^{2}\sum_{m_j=1}^{2}\rho_i(\mathbf{x}_{1n_j}, \mathbf{x}_{2m_j})^2 + \max(0, \gamma_i^{(1,2)} - \rho_i(\mathbf{x}_{1n_j}, \mathbf{x}_{2m_j}))^2 \qquad (6)$$

Each element in the second sum is quadratic function and has its minimum at $\rho_i(\mathbf{x}_{1n_j}, \mathbf{x}_{2m_j}) = \gamma_i^{(1,2)}/2$ and the value of the minimum is $\left(\gamma_i^{(1,2)}\right)^2/2$. So, we can rewrite equation 6 as follows:

$$D_E(\mathbb{Q}_i^{(1)}, \mathbb{Q}_i^{(2)}; \phi_i, j) \geq \frac{\left(\gamma_i^{(1,2)}\right)^2}{2} - \mathcal{L}_{\phi_i}. \qquad (7)$$

Therefore, as the margin $\gamma_i^{(1,2)}$ depends only on the input latent variables and is not trainable, minimizing our embedding loss $\mathcal{L}_{\phi_i}$ maximizes the lower bound for the energy distance $D_E$. This corresponds to learning a Siamese Network $\phi_i(.)$ that separates two probability distributions $\mathbb{Q}_i^{(1)}$ and $\mathbb{Q}_i^{(2)}$, i.e., colored squares and colored diamonds, from each other and minimizes the spread of each distribution, thus resulting in disentangling the effect of $j$ from $i$. The same derivation can be made for the color attribute. After jointly training the Siamese Networks, we can achieve the embedding spaces represented in Figure 3. An example of one such disentangled embedding space is illustrated for MNIST (LeCun & Cortes, 2010) digits in Figure 4 in Appendix B.

## 4 EXPERIMENTS

### 4.1 SETUP

We perform our experiments on a server with Intel Xeon Gold 6134 CPU, 256GB system memory, and an NVIDIA V100 GPU with 16GB of graphics memory. Our generator and discriminator architectures are outlined in our Appendix A. Each knob is a 1-dimensional slice of the latent variable and is sampled from $\text{Unif}(-1, 1)$. We use ADAM (Kingma & Ba, 2014) as an optimizer for our training with the following parameters: learning rate=0.0002 and $\beta_1 = 0.5$. We will release our code after the review process.

**Datasets.** We evaluate our method on two image datasets: (i) the CelebA dataset (Liu et al., 2015), which consists of over 200,000 images of aligned faces. We cropped the images to $64 \times 64$ pixels in size. (ii) the 2D Shapes (Higgins et al., 2017), which is a dataset that is synthetically created with different properties, such as shape, scale, orientation, and x-y locations. Both datasets are divided into training and test sets with a 90%-10% ratio. The weight values for the contrastive loss is $\lambda_\phi = 1$ for the CelebA dataset and $\lambda_\phi = 5$ for the 2D shapes dataset. We use a 32 and 10-dimensional latent variables for the CelebA and the 2D Shapes datasets, respectively.

**Baselines.** We have two versions of our algorithm. UD-GAN refers to the results that are obtained without any guidance at the input of our siamese networks, whereas UD-GAN-G represents a guided training. We compare our method against $\beta$-VAE (Higgins et al., 2017), DIP-VAE (Kumar et al., 2018), and InfoGAN (Chen et al., 2016) to compare against both autoencoder and GAN-based approaches. We get the quantitative and visual results for $\beta$-VAE and DIP-VAE from (Higgins et al., 2017) and (Kumar et al., 2018), and use our own implementation of InfoGAN for training and testing. The same generator/discriminator architecture is used for InfoGAN and our method.

**Guidance.** For the CelebA dataset, the first 28 of 32 latent knobs are unguided and therefore are processed by the same siamese network that outputs a 28-dimensional embedding vector[1]. Whereas the remaining four knobs correspond to four siamese networks ($\phi_{top}, \phi_{miu}, \phi_{mil}, \phi_{bot}$) that are guided with cropped images in Figure 2. For the 2D shapes dataset, we have 10 knobs, where the first 7 dimensions are unguided. In order to guide the remaining three networks, we estimate the center of mass $(\hat{M}_x, \hat{M}_y)$ and the size $\hat{S}$ of the generated object and feed them to our siamese networks, $\phi_X(\hat{M}_x), \phi_Y(\hat{M}_y)$, and $\phi_S(\hat{S})$. More information for this computation can be found in Appendix D.

### 4.2 RESULTS

**Disentanglement Metric.** This metric was proposed by Higgins et al. (2017) and measures whether learned disentangled representations can capture separate semantic variations in a dataset. In $\beta$-VAE and DIP-VAE, this representation is the output of the encoder, i.e., the inferred latent variable. For InfoGAN, we use the representation learned by the discriminator. In our method, we use the concatenated outputs of our siamese networks, which we denote as $\phi(.)$.

The disentanglement metric scores for different methods are illustrated in Table 1. Here, we can see that both of our methods outperforms the baseline on the CelebA dataset. All of the baseline approaches relate the latent variables to generated images on per-image basis. Whereas our approach attempts to relate similarities/differences of latent variable pairs to image pairs, which provides a discriminative image embedding, where each dimension is invariant to unwanted factors (Hadsell et al., 2006).

For both datasets, our guided network (UD-GAN-G) performs better than our unguided approach, especially on the CelebA dataset. This might be due to the correlations between irrelevant attributes. For example the correlation coefficient between "Wearing Lipstick" and "Wavy Hair" attributes is 0.36, although they are not necessarily dependent. One of our guided networks receive the cropped image around the mouth of a person, which prevents cluttering it with hairstyle. Therefore, this guidance provides better disentanglement and results in an improved score as shown in Table 1. Due to containing simple synthetic images, our disentanglement scores for the 2D shapes dataset are very high. The reason we get 100.0 score on our guided method is because of the guidances we choose, which are highly correlated with the ground truth labels, as shown in Table 7 in Appendix D.

---

[1]Note that each dimension is still an independently evaluated

Table 1: Disentanglement metric scores (Higgins et al., 2017), which measure how strongly and independently the dataset attributes are captured by a method.

| Method | 2D Shapes | CelebA |
|---|---|---|
| $\beta$-VAE | 99.2 | 7.1 |
| InfoGAN | 88.4 | 12.3 |
| DIP-VAE | 98.7 | 14.8 |
| UD-GAN | 99.1 | 15.4 |
| UD-GAN-G | **100.0** | **16.5** |

**CelebA Attribute Classification.** Kumar et al. (2018) introduced a binary classification metric for the CelebA attributes that project a test image embedding onto average embedding vectors of attributes. In Table 2, we compare our method against baseline approaches on CelebA attribute classification accuracy using the aforementioned projection vector. Similar to the results in Table 1, our guided approach slightly outperforms our unguided method and the other completely unsupervised techniques. This is because some attributes in the CelebA dataset can be spatially isolated via cropping, which leads to a better classification performance. For example, the attributes that are related to hair (Black Hair, Blond Hair, Wavy Hair) and mouth (Mouth Slightly Open, Wearing Lipstick) are captured better by the guided approach, because our top and bottom crops (see Figure 2) are detaching the effects of other variations and are making attributes less correlated. The accuracy on the attribute "Bangs" is worse on the guided approach. This might be due to heuristic cropping we perform that divides the relevant image region into two slits.

Table 2: CelebA attribute classification accuracy.

| Method | Arched Eyebrows | Attractive | Bangs | Black Hair | Blond Hair | Heavy Makeup | Male | Mouth Slightly Open | No Beard | Wavy Hair | Wearing Hat | Wearing Lipstick |
|---|---|---|---|---|---|---|---|---|---|---|---|---|
| $\beta$-VAE | 71.6 | 72.6 | 90.6 | 79.3 | 89.1 | 79.3 | 83.5 | 76.1 | 86.9 | 67.8 | 95.9 | 82.4 |
| DIP-VAE | 73.7 | 73.2 | 90.9 | 80.6 | 91.9 | 81.5 | 85.9 | 75.9 | 85.3 | 71.5 | 96.2 | 84.7 |
| InfoGAN | 74.7 | 73.8 | **90.9** | 80.9 | 91.5 | 82.6 | 87.4 | 76.9 | **88.7** | 74.3 | **97.3** | 86.9 |
| UD-GAN | **75.0** | 74.8 | 90.5 | 82.1 | 91.5 | **84.2** | 86.2 | 79.7 | 87.5 | 75.1 | 96.5 | 85.6 |
| UD-GAN-G | **75.0** | **75.5** | 90.2 | **82.3** | **92.1** | **84.2** | 89.9 | 82.2 | 87.7 | **75.6** | 96.7 | **87.3** |

**Visual Comparison.** In Table 3, we illustrate images generated by different methods on the CelebA dataset. Each of the three rows capture the change in a semantic property: smile, azimuth, and hair color, respectively. Within each image group, a latent dimension is varied (from top to bottom) to visualize the semantic change in that property. Compared to adversarial methods, such as InfoGAN and UD-GAN-G, the DIP-VAE method generates blurrier images, due to the data likelihood term in VAE-based approaches, which is usually implemented as a pixel-wise image reconstruction loss. In GAN-based approaches, this is handled via a learnable discriminator in an adversarial setting. In Table 1 and 2, we quantitatively show the advantage of using our guided approach. Another advantage is to have better control over the captured attributes. For example, in all unsupervised approaches (including UD-GAN), we need to check which latent dimension represents corresponds to which visual attribute. In some cases, a semantic attribute might not be captured due to the correlated nature of a dataset. Whereas, in UD-GAN-G, we directly obtain the variations in smile, azimuth, and hair color through cropping the bottom, middle, and top part of our images, respectively. Thanks to our guidance in Figure 2, we can directly manipulate these three attributes using the knobs $\mathbf{q}_{bot}$, $\mathbf{q}_{mil}$, and $\mathbf{q}_{top}$ as shown in Table 3.

The same trend is true for the 2D Shapes dataset results in Table 4. Although the X and Y positions and the scale of the synthetic object is captured by both our unsupervised and guided approaches, the guidance we choose directly captures the desired feature on in advance chosen knobs $\mathbf{q}_X$, $\mathbf{q}_Y$, and $\mathbf{q}_S$, respectively.

Table 3: Images generated by varying a latent dimension, which corresponds to a semantic property.

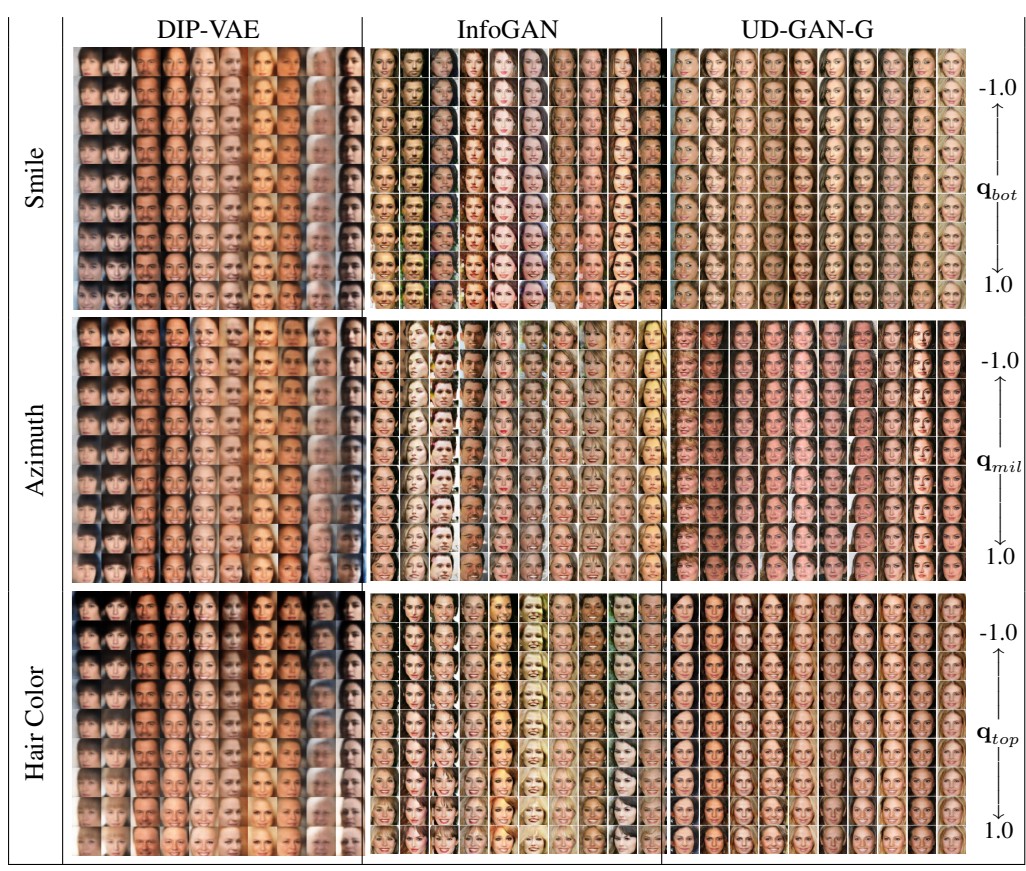

Table 4: Generated images for the 2D Shapes dataset by varying a latent dimension, which corresponds to a semantic property (first row: UD-GAN, second row: UD-GAN-G).

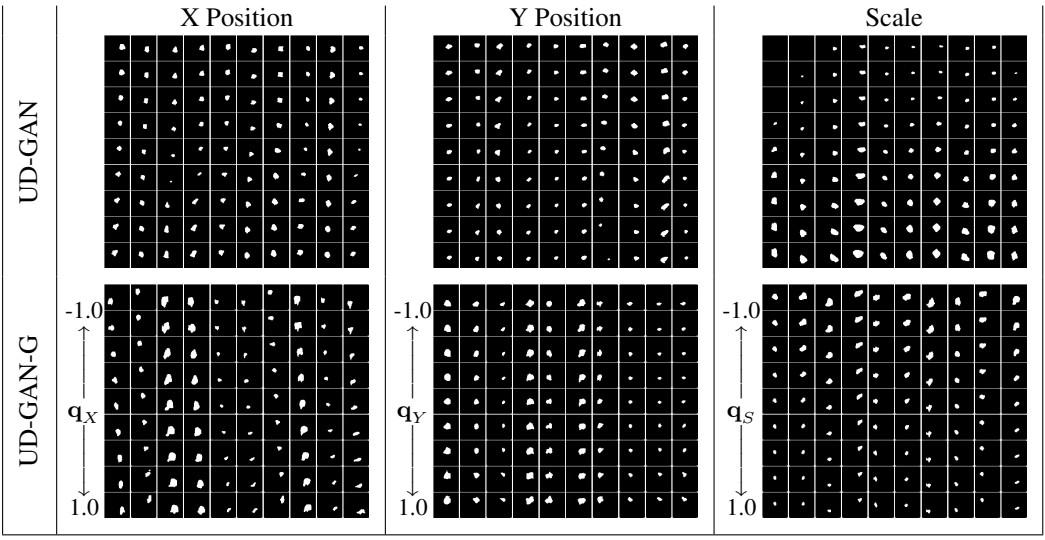

## 4.3 DISCUSSION

In completely unsupervised approaches, there is no guarantee to capture all of the desired semantic variations. The main premise behind UD-GAN-G is to find very simple, yet effective ways to

capture some of the variation in the data. This weak supervision helps us to obtain proxies to certain semantic properties, so that we get the desired features without training the model multiple times with different hyperparameters or initializations.

In the aligned the CelebA dataset, each face is roughly centered around the nose. This reduces the variation and simplifies the problem of guidance design, as we show in Figure 2. In more complex scenarios, where the objects can appear in a large variety of scales, translations, and viewpoints, one can use a pre-trained object detection and localization method, such as YOLO (Redmon et al., 2015), as a guidance network. This enables us to use the knowledge obtained from a labeled dataset, such as ImageNet (Russakovsky et al., 2015) to disentangle a new unlabeled dataset. Note that backpropagating the gradients of a deep network into an image might cause adversarial samples (Szegedy et al., 2014). However, the discriminator can alleviate this by rejecting problematic images.

In order to backpropagate the gradients from the siamese networks to the generator, the guidance function we use needs to be differentiable. This might pose a limitation to our method; however, differentiable relaxations can instead be used to guide our network. For example, one can employ differentiable relaxation of the superpixel segmentation in (Jampani et al., 2018) to disentangle a low-level image segmentation.

Our latent variables are sampled from a uniform distribution. In addition, image similarity is measured by using L2-distance between a pair of image embeddings. We experimented with modeling some latent dimensions as categorical variables. However, we encountered training stability issues, due to computing the softmax loss between two learnable categorical image embeddings, instead of one embedding and one fixed label vector as it is usually done. We plan to tackle that problem in our future work.

## 5 CONCLUSION

In this paper we introduced UD-GAN and UD-GAN-G, novel GAN formulations which employ Siamese networks with contrastive losses in order to make slices of the latent noise space disentangled and more semantically meaningful. Our experiments encompassed guided and unguided approaches for the embedding networks, and illustrated how our methods can be used for semantically meaningful image manipulation. Our qualitative and quantiative results confirm that our method can adjust well to the intrinsic factors of variation of the data and outperform the current state-of-the-art methods on the CelebA and 2D Shapes datasets. In future work, we plan to investigate more powerful forms of embedders, e.g. extracting information from pre-trained networks for semantic segmentation and landmark detection. This allows for even more powerful novel image manipulation techniques.

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

APPENDIX

## A  NEURAL NETWORK ARCHITECTURES

In Table 5, we show the neural network layers we use in our generator for different datasets. Our discriminator and siamese network architectures are the inverted version of our generator. Each fully connected and Conv2D layer is followed by a Leaky ReLU non-linearity, except the last layer.

Table 5: The architectures of our generator networks.

| Layer | CelebA | 2D Shapes |
|---|---|---|
| Latents | (32) | (10) |
| Fully Connected | (2048) | (512) |
| Reshape | $(128 \times 4 \times 4)$ | $(32 \times 4 \times 4)$ |
| Conv2D-Transpose $(3 \times 3)$ | $(128 \times 8 \times 8)$ | $(32 \times 8 \times 8)$ |
| Conv2D $(3 \times 3)$ | $(128 \times 8 \times 8)$ | $(32 \times 8 \times 8)$ |
| Conv2D-Transpose $(3 \times 3)$ | $(128 \times 16 \times 16)$ | $(32 \times 16 \times 16)$ |
| Conv2D $(3 \times 3)$ | $(128 \times 16 \times 16)$ | $(32 \times 16 \times 16)$ |
| Conv2D-Transpose $(3 \times 3)$ | $(128 \times 32 \times 32)$ | $(32 \times 32 \times 32)$ |
| Conv2D $(3 \times 3)$ | $(128 \times 32 \times 32)$ | $(32 \times 32 \times 32)$ |
| Conv2D-Transpose $(3 \times 3)$ | $(128 \times 64 \times 64)$ | $(32 \times 64 \times 64)$ |
| Conv2D $(3 \times 3)$ | $(128 \times 64 \times 64)$ | $(32 \times 64 \times 64)$ |
| Conv2D $(1 \times 1)$ | $(3 \times 64 \times 64)$ | $(1 \times 64 \times 64)$ |

## B  CHOOSING SEMANTICS WITH GUIDANCE

The Siamese Networks $\phi_i$ are desired to map images into embedding spaces, where they can be grouped within a distinct semantic context. For the example shown in Figure 4, where we disentangle the shape and the color, this might not be directly achievable in a completely unsupervised setting, because the separation in equation 4 is not unique. However, we can still benefit from the disentangling capability of our method via small assumptions and domain knowledge, without collecting labeled data.

Consider the toy example, where we extend the MNIST dataset (LeCun & Cortes, 2010) to have a random color, sampled from a uniform RGB color distribution. We define our problem to independently capture the shape of a digit with $\mathbf{q}_1$ and its color with $\mathbf{q}_2$.

In Figure 4(a), we show images created by a generator, which is trained along with two networks, $\phi_1$ and $\phi_2$, without any guidance in an unsupervised setting. We can see that the knobs, $\mathbf{q}_1$ and $\mathbf{q}_2$, capture the variations in the data, however, these variations are coupled with multiple semantic properties. Each knob modifies a complicated combination of shape and color.

However, if we *design* a network architecture in a *slightly smarter* way, we should be able to separate the shape and the color attributes. This is exemplified in Figure 4(b), where instead of feeding the whole image to $\phi_2$, we feed the average color of some randomly sampled pixels from a generated image. This choice prevents $\phi_2$ to capture the spatial structure of the generated digit and to focus only on color. After the training our method with a modified $\phi_2$, the first network captures shape of a digit, and the second one captures the color variations. This can also be observed in Figure 4(c) and 4(d), where we use t-SNE (van der Maaten & Hinton, 2008) to visualize embedding spaces for shape and color, respectively.

## C  EXPERIMENTS ON ADDITIONAL GUIDANCES

In order to show the effect of the guided siamese networks, we perform three experiments on the MS-Celeb dataset (Guo et al., 2016) by using different guiding proxies. In the first experiment, only one of the two networks is guided with an edge detector at the input. Results of this experiment are shown in Table 6. We can see that the first knob, which is connected to edges, captures the overall

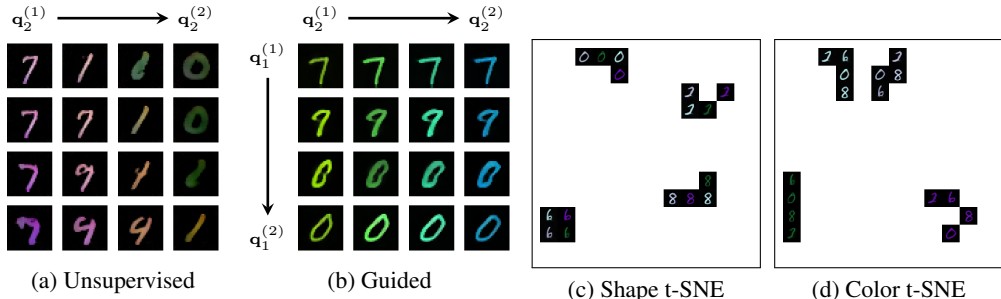

(a) Unsupervised      (b) Guided      (c) Shape t-SNE      (d) Color t-SNE

Figure 4: (a) Samples from the colored version of the MNIST dataset. (b) Images generated after an unsupervised training with two knobs and (c) after a guided training (the knob values are interpolated between two values and then concatenated to generate the final image). (d) The t-SNE representation of the embedding vectors for shape and (e) color.

outline and roughly controls the identity of the generated face. On the other hand, the unguided second knob modifies the image with minimal changes to image edges. This change, in this case, corresponds to the lighting of the face.

We perform a second experiment with the edge detector, where in this case, the second knob is guided with the average color of the generated image. In Table 6, we can observe the results of our disentangled image manipulation. The first knob with the edge detector again captures the outline of the face, and the second average color knob modifies a combination of the light and the skin color, similar to the results in Figure Table 6.

In our third experiment, we employ the cropped guidance networks. The two knobs receive the cropped top and bottom part of the image for training. Although these image crops are not independent, we still get acceptable results that are shown in Table 6. Adjusting the first knob only modifies the upper part of the face; the hair and the eyes. Similarly, the second knob is responsible for determining the chin and mouth shape.

Table 6: The results of UD-GAN-G using differently guided siamese networks.

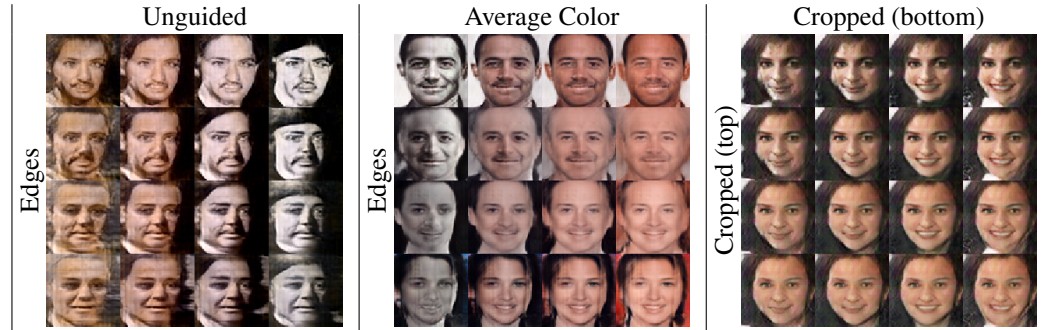

## D GUIDING FOR THE 2D SHAPES DATASET

In order to guide our siamese networks for the 2D shapes dataset, we estimate the center of mass of the generated image, and the size of the generated object as follows:

$$\hat{M}_x = \frac{1}{Z} \sum_{c_x, c_y} c_x \cdot \mathbf{x}[c_x, c_y], \qquad \hat{M}_y = \frac{1}{Z} \sum_{c_x, c_y} c_y \cdot \mathbf{x}[c_x, c_y]$$

$$\hat{S} = \frac{1}{Z} \sum_{c_x, c_y} \left( (c_x - \hat{M}_x)^2 + (c_y - \hat{M}_y)^2 \right) \cdot \mathbf{x}[c_x, c_y]$$

$$Z = \sum_{c_x, c_y} \mathbf{x}[c_x, c_y], \tag{8}$$

where, $\mathbf{x}$ is a generated image, $\mathbf{x}[c_x, c_y]$ is the pixel intensity at image coordinates $[c_x, c_y]$, $(\hat{M}_x, \hat{M}_y)$ are the coordinates of the center of mass of $\mathbf{x}$, and $\hat{S}$ is the size estimate for the generated object. As the 2D shapes dataset is relatively simple and contain only one object, these guidances are highly correlated with the ground truth attributes as shown in Table 7.

Table 7: Correlation between ground truth attributes of the 2D shapes dataset and the calculated proxies.

| Ground Truth Attribute | $\hat{M}_x$ | $\hat{M}_y$ | $\hat{S}$ |
|---|---|---|---|
| Shape | 0.000 | -0.002 | -0.366 |
| Scale | -0.000 | -0.000 | 0.910 |
| Orientation | 0.027 | 0.000 | -0.001 |
| X Position | 0.998 | -0.000 | -0.000 |
| Y Position | -0.000 | 0.998 | 0.001 |

## E ADDITIONAL SEMANTIC MANIPULATION

In Figure 5, we illustrate additional semantic properties that are captured by UD-GAN-G.

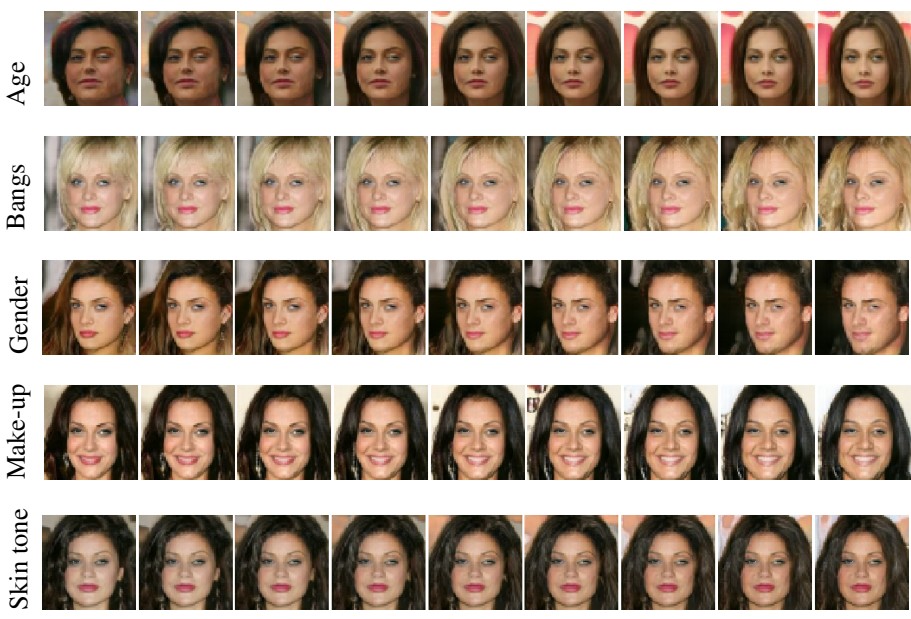

Figure 5: Semantic properties that are captured by our method.

# F CELEBA ATTRIBUTE CLASSIFICATION

In Table 8, we compare the classification perfromance of our method to InfoGAN on all attributes in the CelebA dataset.

Table 8: CelebA attribute classification accuracy.

| Method | 5 o Clock Shadow | Arched Eyebrows | Attractive | Bags Under Eyes | Bald | Bangs | Big Lips | Big Nose | Black Hair | Blond Hair | Blurry | Brown Hair | Bushy Eyebrows | Chubby | Double Chin | Eyeglasses | Goatee | Gray Hair | Heavy Makeup | High Cheekbones |
|---|---|---|---|---|---|---|---|---|---|---|---|---|---|---|---|---|---|---|---|---|
| InfoGAN | **90.4** | 74.7 | 73.8 | 80.6 | **97.9** | **90.9** | 67.5 | **80.7** | 80.9 | 91.5 | 94.9 | **82.1** | 87.8 | **94.6** | 95.4 | 94.5 | **95.8** | **97.0** | 82.6 | 78.0 |
| UD-GAN-G | **90.4** | **75.0** | **75.5** | **81.3** | **97.9** | 90.2 | **68.1** | 80.7 | **82.3** | **92.1** | **95.1** | 81.9 | **88.6** | **94.6** | **95.5** | **95.4** | 95.5 | **97.0** | **84.2** | **81.2** |

| | Male | Mouth Slightly Open | Mustache | Narrow Eyes | No Beard | Oval Face | Pale Skin | Pointy Nose | Receding Hairline | Rosy Cheeks | Sideburns | Smiling | Straight Hair | Wavy Hair | Wearing Earrings | Wearing Hat | Wearing Lipstick | Wearing Necklace | Wearing Necktie | Young |
|---|---|---|---|---|---|---|---|---|---|---|---|---|---|---|---|---|---|---|---|---|
| InfoGAN | 87.4 | 76.9 | **96.1** | **85.3** | **88.7** | 72.0 | 95.9 | 72.9 | 91.5 | 93.2 | **95.6** | 82.1 | 79.1 | 74.3 | 79.8 | **97.3** | 86.9 | **86.3** | **93.0** | 80.9 |
| UD-GAN-G | **89.9** | **82.2** | **96.1** | 85.2 | 87.7 | **72.4** | **96.4** | **73.6** | **92.0** | **93.9** | 95.3 | **86.4** | **79.2** | **75.6** | **80.4** | 96.7 | **87.3** | **86.3** | **93.0** | **81.1** |

# G ATTRIBUTE CORRELATIONS.

In Table 9, we compare the correlation between different embedding (or latent) dimensions and the correlation between embedding dimensions and the CelebA attributes. Although DIP-VAE encodes a more un-correlated representation, due to the correlated nature of CelebA attributes, it does not necessarily transfer to a disentangled semantic representation, as illustrated by the quantitative results in Table 1 and 2.

Table 9: Correlation between embeddings (or latents) with each other (first row) and with CelebA attributes(second row). Negative correlations are inverted for visibility purposes.

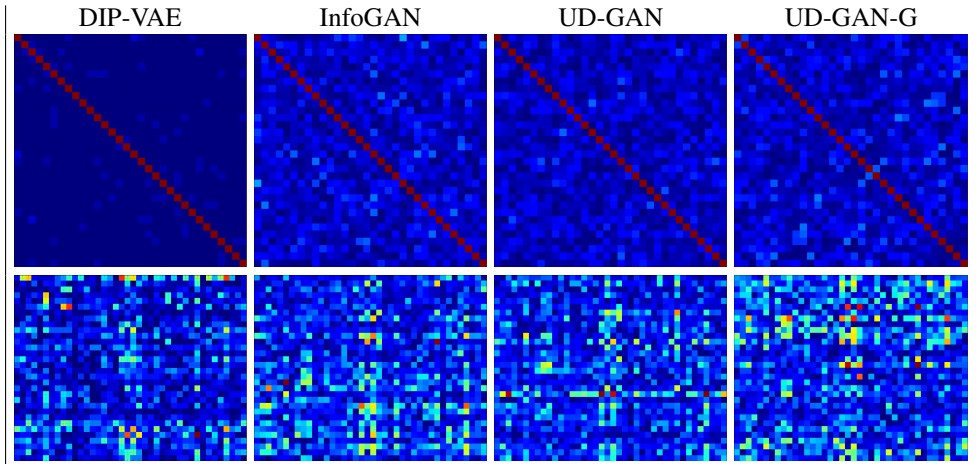

