# OpenReview forum: "Unlabeled Disentangling of GANs with Guided Siamese Networks"
_ICLR.cc/2019/Conference_

### Official Review · AnonReviewer1 · 2018-10-30
**Interesting problem but limited approach and evaluation**

**Rating:** 6
**Confidence:** 4

**Review:**

Summary

The paper presents a novel approach for learning a generative model where different factors of variations can be independently manipulated. The method is build upon  the GAN framework where the latent variables are divided into different subsets (chunks) which are expected to encode information about high-level factors of variation. To this end, a Siamese Network for each chunk is trained with a contrastive loss minimizing the distance between generated images sharing the same factor (the latent variables in the chunk are equal), and maximizing the distance between pairs where the latent variables differ. Given that the proposed model fails in this fully-unsupervised setting, the authors propose to add weak-supervision into the model by forcing the Siamese networks to  focus only on particular aspects of generated images (e.g, color, edges, etc..). This is achieved by applying  a basic transformation  over the input images in order to remove specific information. The evaluation of the  proposed model is carried out using the MS-Celeb dataset where the authors provide qualitative results.


Methodology

*Disentangling generative factors without explicit labels is a challenging and interesting problem. The idea of dividing the latent representation in different subsets and using a proxy task involving triplets of images has been already explored in [3]. However, the use of Siamese networks in this context is novel and sound.

*As shown in the reported results, the proposed method fails to learn meaningful factors in the unsupervised setting. However, the authors do not provide an in-depth discussion of this phenomena. Given that previous works [1,2,3] have successfully addressed this problem using a completely unsupervised approach, it would be necessary to give more insights about: (i) why the proposed method is failing (ii) why this negative result is interesting and (iii) if the method could be useful in other potential scenarios.

*The strategy proposed to introduce weak-supervision is too ad-hoc. I agree that using cues such as the average color of an image can be useful if we want to model basic factors of variation. However, it is unclear how a similar strategy could be applied if we are interested in learning variables with higher-level semantics such as the expression of a face or its pose.

*As far as I understand, the transformations applied to the input images (e.g, edge detection) must be differentiable (given that it is necessary to backpropagate the gradient of the contrastive loss through the generator network). If this is the case, this should be properly discussed in the paper. Moreover, given that the amount of differentiable transformations is reduced, this also limits the application of the proposed method for more interesting scenarios.

*It is not clear why the latent variables modelling the generative factors are defined using a Gaussian prior. How the case where two images have a very similar latent factor is avoided while generating pairs of images for the Siamese network?  Have the authors considered to use categorical or binary variables? The use of the contrastive loss sounds more appropriate in this case.


Experimental results

*The experimental section is too limited. First of all, only a small number of qualitative results are reported and, therefore, it is very difficult to assess the proposed method and draw any conclusion. For example, when the edge extractor is used, what kind of information is modeled by the latent variables? Is it consistent across different samples?

Moreover, it is not clear why the authors have limited the evaluation to the case where only two “chunks” are used. In principle, the method could be applied with many more subsets of latent variables and then manually inspect them to check it they are semantically meaningful (see [2])

*As previously mentioned, there are many recent works addressing the same problem from a fully-unsupervised perspective [1,2,3]. All these works provide quantitative results evaluating the learned representations by using them to predict real labels (e.g, attributes in the CelebA data-set). The authors could provide a similar evaluation for their method by using the feature representations learned by the siamese networks in order to evaluate how much information they convey about real factors of variation. This could clarify the advantages of the weakly-supervised strategy compared to unsupervised approaches.

Review summary

+The addressed problem (learning disentangled representations without explicit labeling) is challenging and interesting.

+The idea of using a proxy task (contrastive loss with triplets of generated images) is somewhat novel and promising.

- The authors report only negative results for the fully-unsupervised version of UD-GAN The paper lacks and in-depth discussion about why this negative result is interesting.

-The strategy proposed to provide weak-supervision to the model is too ad-hoc and it is not clear how to apply it in general applications

-The experimental section do not clarify the benefits of the proposed approach. In particular, the qualitative results are too limited and no quantitative evaluations is provided.


[1] Variational Inference of Disentangled Latent Concepts from Unlabelled Observations (Kumar et al, ICLR 2018)

[2] Beta-vae: Learning basic visual concepts with a constrained variational framework. (Higgins et. al, ICLR 2017)

[3] Disentangling Factors of Variation by Mixing Them. (Hu et. al, CVPR  2018)

---

> ### Author Response · Authors · 2018-11-23
> **Author Response**
>
> We would like to thank you for reviewing our paper.
>
> [Unguided Case] Please refer to our general comment above on why our unguided case performs better now. The main usefulness of our guided approach is to directly capture some of the desired variations in the data. This is now clearer on our quantitative and visual results in the “Experiments” section.
>
> [Heuristic Guidance] The main premise behind guiding our siamese networks is to find very simple, yet effective ways to capture some of the variation in the data, through weak supervision. For more complex semantics, we discuss the possibility of using a pre-trained network as guidance. Please refer to our “Discussion” section for more details.
>
> [Differentiable Guidance]  The transformations need to be differentiable in order to backpropagate the gradients into our generator. This is now pointed out and discussed in our "Discussion" section. Although this limits the function families, we can still use differentiable relaxations of more complicated functions.
>
> [Gaussian Prior on Latents] In our new experiments, we used uniform distributions to model the generative factors. We had experiments with categorical variables, however, we faced training stability issues with them. We now point this out in our "Discussion" section.
>
> [Similar Latent Factors] We now use an adaptive margin that depends on the distance between two latent samples. So, if samples are close to each other, the margin is smaller, and vice versa.
>
> [Experiments Section] We now compare our method against Beta-VAE, DIP-VAE, and InfoGAN, both qualitatively and quantitatively. Please refer to our updated "Experiments" section.
>
> [Information of Guidance] In Figure 3, we visualize which part of an image was visible to a siamese network. In addition, we show how changing the corresponding guided knob affects the generated images.
>
> [More Than Two Attributes] We now use 32 dimensions for the CelebA dataset and 10 dimensions for the 2D shapes dataset.

---

> > ### Comment · AnonReviewer1 · 2018-11-28
> > **feedback on author response**
> >
> > Dear authors,
> >
> >
> > I appreciate the efforts made during the rebuttal period. I think that the quality of the paper has been significantly increased. Specially, the inclusion of the soft-margin term in the contrastive loss sounds very interesting.
> >
> >
> > Although some of my initial concerns have been addressed, there are some remaining issues which still remain unclear. Moreover, I have new questions given the large amount of new experimental results added.
> >
> >
> > [Unguided Method]
> >
> > By comparing UD-GAN with state-of-the-art methods, you have shown that you are able to outperform other unsupervised approaches in standard benchmarks. However, I think that the followed experimental setup is not clearly explained in the paper:
> >
> >
> > -How many “knobs”/Siamese networks are used in UD-GAN? If it’s only one (as I have understood), how is the method supposed to model different variation factors?
> >
> >
> > -What is the dimensionality of the latent representation used to compute the disentanglement metric?  Is it the same than the one used in the compared methods? Otherwise, the reported results are not directly comparable.
> >
> >
> > -The authors do not show any qualitative result on the CelebA dataset for the unguided version. Does UD-GAN learn semantically meaningful factors in these dataset?
> >
> >
> > Without all this information it is difficult to assess that UD-GAN is really learning disentangled representations. Moreover, as the authors state in the paper, there is not any guarantee that their unsupervised method will learn to model high-level generative factors. Therefore, it is counter-intuitive that UD-GAN outperformed all the previous state-of-the-art unsupervised methods. The paper does not provide any convincing explanation or discussion about this issue.
> >
> >
> >
> > [Guided Method]
> >
> > I still think that the approach used to provide weak-supervision is too ad hoc. I am not convinced about how the proposed strategy can be applied to real scenarios where removing information related with the generative factors can be extremely difficult. For example, in the CelebA dataset, how the followed strategy could be used to disentangle the “Make-up” attribute?
> >
> >
> > Apart from this, I have also questions about the experimental setup followed to evaluate the guided method. In particular, it is unclear which is the representation used to compute the disentangle metric. As far as I have understood, it is the concatenation of the last layer of the different Siamese networks. This results in a much larger dimensionality compared to the one used in UD-GAN and in the compared baselines. As a consequence, the reported numbers are not directly comparable.
> >
> >
> > Also related with this issue, if one Siamese network is supposed to capture information about one high-level factor (e.g, hair color), why the representation of all the Siamese networks is used?. In my opinion, a convincing evaluation would consists on using only the representation of the Siamese network that is supposed to model the specific attribute  (e.g \psi_{top}).  This is the only way to actually show that the proposed method is disentangling the different high.level attributes.
> >
> >
> > [Revised score]
> >
> > In conclusion, I slightly updated my score given the additional material provided in the updated version. However, I still think that the paper is not ready for publication given all the discussed issues.

---

> > > ### Author Response · Authors · 2018-12-03
> > > **Author Response**
> > >
> > > Dear Reviewer,
> > >
> > > Thank you again for your review and inquires. You can find our response below.
> > >
> > > [Embedding Dimensionality or Our Methods]
> > >
> > > In order to compute the disentanglement score, we chose our embedding dimensionality to be consistent with Beta-VAE and DIP-VAE. For the CelebA dataset, our embedding vector has a total of 32 dimensions. For the 2D shapes dataset, it has 10 dimensions. We take these values from Beta-VAE, which are also repeated by DIP-VAE. You can find more details for our unsupervised and guided approaches below. We will explicitly emphasize this in the next iteration of our paper.
> > >
> > > (UD-GAN) For the CelebA dataset, we have 32 knobs, each correspond to a one-dimensional latent slice. Our generator maps this 32-dimensional latent vector into images. After that, we could follow two ways to embed a generated image:
> > >
> > > 1) By using 32 different siamese networks, each produce a 1-dimensional embedding
> > > 2) By using 1 siamese network, which produces a 32-dimensional embedding
> > >
> > > We use the second approach, which shares one siamese network and embeds an image into a 32-dimensional embedding vector. The main reason for this parameter sharing is to save GPU memory. Note that, for each knob, we calculate the contrastive loss by only using the corresponding single embedding dimension, not on the whole 32-dim embedding vector. This means that, each latent dimension corresponds to a single and unique embedding dimension.
> > >
> > > (UD-GAN-G) For the CelebA dataset, we use 32 knobs. The first 28 knobs are unguided and processed by the same siamese network, which embeds a generated image into a 28-dimensional embedding vector. As we explained for the unguided case above, this is only to save GPU memory and each embedding dimension is still separately treated. For the remaining 4 knobs, a generated image is guided by four image crops and fed into four different siamese networks, each embeds an image into a 1-dimensional vector. The concatenation of all siamese network outputs has thus 32-dimensions. For the 2D shapes dataset, we use 10 knobs. The first 7 knobs are unguided and the rest three knobs are guided with three separate siamese networks.
> > >
> > > [Visual Results for UD-GAN]
> > >
> > > We did not include visual results for UD-GAN case due to space constraints. It indeed learns semantically meaningful factors. We will add these visual results to our Appendix for a better comparison.
> > >
> > > [UD-GAN High-Level Generative Factors and Performance]
> > >
> > > In Section “3.4 Probabilistic Interpretation”, we show that minimizing the contrastive loss results in a disentangled embedding representation. This is also illustrated in the correlation matrix of the inferred embedding vectors in Table 9 in Appendix G. Some of the embedding dimensions are strongly related to certain high-level generative factors, because it was possible to use our embedding vectors to classify the existence of an attribute, as shown in Table 2. With our unsupervised method (UD-GAN), there is no guarantee to capture all of the desired high-level generative factors. However, it is still possible to model some of them, as long as the GAN does not have convergence issues, such as mode collapse.
> > >
> > > [Siamese Networks - Hair Color Attribute - Disentangling]
> > >
> > > In Figure 2, for the hair color attribute, we only change the knob \q_{top} that corresponds to the guided siamese network \phi_{top}. The other latent dimension values are kept the same. The output of \phi_{top} is a one-dimensional embedding, but is not shown in Figure 2.
> > >
> > > In Table 9, Appendix G, we show the correlation matrix of the concatenated embedding vectors (in total 32 dimensions for the CelebA dataset) both for UD-GAN and UD-GAN-G. We infer embedding vectors by passing real images through our siamese embedding networks and (similar to DIP-VAE) computing the correlation matrix on these vectors. As illustrated, individual embedding dimensions are uncorrelated with each other and correlated with real high-level attributes in the CelebA dataset.
> > >
> > > [Ad-Hoc Guidance in UD-GAN-G]
> > >
> > > Coming up with a guidance function is easier for certain variations than others. Our main goal is not to design guidance functions for each and every attribute. Instead, our guidance approach complements the literature in the sense that it offers a way to disentangle some of the spuriously correlated variations without labeled data. The rest of the variations can be modeled in an unsupervised way, similar to how UD-GAN and other unsupervised techniques operate.

---

> > > > ### Comment · AnonReviewer1 · 2018-12-10
> > > > **feedback on author response**
> > > >
> > > > Dear Authors,
> > > >
> > > > Thank you for your response. The experimental setup makes more sense to me after your clarifications As a consequence, I have increased my score from 5 to 6. However, all the explanations should be properly discussed in the paper.
> > > >
> > > > I can not give a higher score  given that I still think that the proposed approach to add weak-supervision is too ad-hoc and difficult to apply in real scenarios. On the positive side, the unsupervised version of the model has its own merit.

---

### Official Review · AnonReviewer2 · 2018-11-02
**Very interesting idea with insufficient experimental validation**

**Rating:** 5
**Confidence:** 4

**Review:**

The paper proposes a framework for learning interpretable latent representations for GANs. The key idea is to use siamese networks with contrastive loss. Specifically, it decomposes the latent code to a set of knobs (sub part of the latent code). Each time it renders different images with different configurations of the knobs. For example, 1) as changing one knob while keeping the others, it expects it would only result in change of one attribute in the image, and 2) as keeping one knob while changing all the others, it expects it would result in large change of image appearances. The relative magnitude of change for 1) and 2) justifies the use of a Siamese network in addition to the image discriminator in the standard GAN framework. The paper further talks about how to use inductive bias to design the Siamese network so that it can control the semantic meaning of a particular knob.

While I do like the idea, I think the paper is still in the early stage. First of all, the paper does not include any numerical evaluation. It only shows a couple of examples. It is unclear how well the proposed method works in general. In addition, the InfoGAN work is designed  for the same functionality. The paper should compare the proposed work to the InfoGAN work both quantitatively and qualitatively to justify its novelty.

---

> ### Author Response · Authors · 2018-11-23
> **Author Response**
>
> We would like to thank you for reviewing our paper.
>
> [Experiments Section] We have significantly updated qualitative and quantitative results in our "Experiments" section and now compare our methods against Beta-VAE, DIP-VAE, and InfoGAN.
>
> [InfoGAN] Compared to InfoGAN, our method is novel in two ways: First, we use separate networks to obtain the image embeddings, which enables us to guide some of these networks with simple functions. The guidance allows more control over the latent space, even in lack of data. Second, we use pairwise similarity/dissimilarity in order to perform disentangling, which is different from InfoGAN's approach of maximizing the label likelihood. This point is now addressed in our "Related Work" section.

---

### Official Review · AnonReviewer4 · 2018-11-12

**Rating:** 6
**Confidence:** 3

**Review:**

[EDIT]: I have updated my score after the author response and paper revision.
=============================

[I was asked to step in as a reviewer last minute. I did not look at the other reviews].

-------------------------------
Summary
-------------------------------
This paper proposes to learn disentangled latent states under the GAN framework. The core idea is to partition the latent states into N partitions, and correspondly have N Siamese networks that pull the generated images with the same latent partition towards each other, along with a contrastive loss which ensures generated images with different latent partitions to be different. The authors experiment with two setups: in the "unguided setup" training is completely unsupervised, while in the "guided" setup, there is some weak supervision to encourage different partitions to learn different factors.

-------------------------------
Evaluation
-------------------------------
While the motivation is nice, I find the results (especially in the unguided setup) underwhelming. This does not seem surprising to me, as in the unguided case, the constrative loss seems not strong enough to encourage the latent partitions to be different. Results with weak supervision (their method for injecting weak supervision was very nice) are more impressive. However, there is no comparison against existing work. Learning disentangled representations with deep generative models is very much an active area. Here are some recent papers:

https://openreview.net/references/pdf?id=Sy2fzU9gl
https://arxiv.org/abs/1802.05822
https://arxiv.org/abs/1802.05983
https://arxiv.org/abs/1802.04942

Importantly, there are no quantitative metrics. I do not think this work is ready for publication.

---

> ### Author Response · Authors · 2018-11-23
> **Author Response**
>
> We would like to thank you for reviewing our paper.
>
> [Unguided Case and Disentanglement] Please refer to our general comment above on why our unguided case performs better now. We also updated our “Probabilistic Interpretation” section with analysis on how the contrastive loss helps us to learn a disentangled representation. Evidence and comparison to other methods on disentanglement is provided in  Table 9 in Appendix G, where we visualize the correlations between our embedding dimensions.
>
> [Experiments Section] We have significantly updated qualitative and quantitative results in our "Experiments" section and now compare our methods against Beta-VAE, DIP-VAE, and InfoGAN.

---

> > ### Comment · AnonReviewer4 · 2018-11-24
> > **response**
> >
> > Thank you for the updated paper. The revised version is significantly better than the initial submission and addresses many of the points raised (most importantly, it provides quantitative comparison against existing methods).  I have updated my score based on the latest iteration of the paper.

---

> > > ### Author Response · Authors · 2018-12-03
> > > **Author Response**
> > >
> > > Thank you for your response. Please do let us know if you have any further inquires about the updated version of our paper.

---

### Official Review · AnonReviewer5 · 2018-11-13
**Interesting idea but incomplete justifications**

**Rating:** 5
**Confidence:** 4

**Review:**

[Edit] I changed my rating from 4 to 5 based on the author responses.
=======
This paper proposed a GAN that learns a disentangled factors of variations in unsupervised (or weakly-supervised) manner. To this end, the proposed method incorporates a contrastive loss together with Siamese network, which encourages the generator to output smaller variations in samples if they are drawn by varying the same latent factors. The proposed idea is evaluated on simple datasets such as MNIST and centered faces, and show that it is able to learn disentangled latent codes by incorporating some heuristics.

Although the paper presents an interesting and reasonable idea, I think the paper is incomplete and in the proof-of-concept stage. In terms of method, the guidance for learning Siamese networks are designed heuristically (e.g. edges, colors, etc.) which limits its applicability over various datasets; I think that designing more principled approach to build such guidances from data should be one of the key contributions of the paper. In terms of evaluation, the authors only presented a few qualitative results on simple datasets, which is not comprehensive and convincing.

In conclusion, I suggest a reject of this paper due to the lacks of comprehensive study and evaluation.

---

> ### Author Response · Authors · 2018-11-23
> **Author Response**
>
> We would like to thank you for reviewing our paper.
>
> [Principled Guidance] The design of guidances is heuristic, but as illustrated in Figure 2 and in Table 2, they are easy to design and are effective. Further, we added our unsupervised analyses to show that the method works even without explicit guidance on all tested datasets. In this paper, we propose the idea of guidance itself and show that it is imposing a desired semantics on the latent space without having labeled data. In our future work, we plan to investigate more principled ways of deciding guidances. We now address this point in our "Discussion" section.
>
> [Experiments Section] We have significantly updated qualitative and quantitative results in our "Experiments" section and now compare our methods against Beta-VAE, DIP-VAE, and InfoGAN.

---

> > ### Comment · AnonReviewer5 · 2018-12-10
> > **Rebuttal response**
> >
> > I appreciate the authors' efforts made during the rebuttal period. The new results, especially the comparisons against other methods and the experiments on unguided approach, made the paper much stronger. However, I still have some concerns about the practical usefulness of the guidance employed in the paper, as it is designed heuristically specifically suitable for a certain dataset. Considering the main contribution of the paper is introducing a generative framework that can incorporate the additional guidance to learn disentangled representation, demonstrating the results with only ad-hoc guidance on a few specific datasets looks considerable drawback to me to recommend the acceptance of the paper.

---

> > > ### Author Response · Authors · 2018-12-14
> > > **Author Response**
> > >
> > > Thank you again for your review.
> > >
> > > UD-GAN and other unsupervised techniques already capture and disentangle various attributes in a given dataset. In this paper, our guided approach (UD-GAN-G) complements the unsupervised literature by offering a simple way to further disentangle some of the spuriously correlated variations without labeled data. For some datasets and desired behaviours, the design of a guidance function is straightforward - while for others, as correctly pointed out, it can be a very difficult task. However, our main goal is not to design guidance functions for each and every attribute. Instead, we supplement the unsupervised disentanglement process with guidance.

---

### Author Response · Authors · 2018-11-23
**Changes in the Paper**

We would like to thank all of our reviewers for their insightful comments. Inspired by their suggestions, we have performed the following changes in our paper:

- We updated our abstract to be more consistent with the changes in our paper
- We changed the loss function we use from the WGAN-GP [1] to the original GAN loss in [2]. This significantly helped our approach to disentangle without guidance.
- We have empirically found out that the gradient penalty term in WGAN-GP loss was preventing our unsupervised method to learn a disentangled representation. However, a theoretical insight on why this happens requires further analysis.
- We added a section to explain guidance functions and their purpose.
- We updated our “Probabilistic Interpretation” section to be more concise.
- We significantly updated our “Experiments” section with quantitative and qualitative comparisons with the state-of-the-art techniques, such as Beta-VAE [3], DIP-VAE [4], and InfoGAN [5].
- We improved our “Discussion” section to address the limitations of our method.

[1] Improved Training of Wasserstein GANs (Gulrajani et. al., NIPS 2017)
[2] Generative Adversarial Networks (Goodfellow et. al., NIPS 2014)
[3] Beta-vae: Learning basic visual concepts with a constrained variational framework. (Higgins et. al., ICLR 2017)
[4] Variational inference of disentangled latent concepts from unlabeled observations. (Kumar et. al., ICLR 2018)
[5] Infogan: Interpretable representation learning by information maximizing generative adversarial nets. (Chen et. al., NIPS, 2016)

---

### Meta-Review · Area_Chair1 · 2018-12-11

**Confidence:** 4
**Recommendation:** Reject

**Metareview:**

The paper received mixed reviews. It proposes a variant of Siamese network objective function, which is interesting. However, it’s unclear if the performance of the unguided method is much better than other baselines (e.g., InfoGAN). The guided version of the method seems to require much domain-specific knowledge and design of the feature function, which makes the paper difficult to apply to broader cases.